# Human muscle activity and lower limb biomechanics of overground walking at varying levels of simulated reduced gravity and gait speeds

**Mhairi K. MacLean**¤*, **Daniel P. Ferris**[ID]*

J. Crayton Pruitt Family Department of Biomedical Engineering, University of Florida, Gainesville, Florida, United States of America

¤ Current address: Department of Mechanical Engineering, The City College of New York, New York, New York, United States of America
* mmaclean@ccny.cuny.edu (MKM); dferris@bme.ufl.edu (DPF)

**Data Availability Statement:** The data supporting the findings of this study are openly available in Figshare (http://figshare.com/collections/

## Abstract

Reducing the mechanical load on the human body through simulated reduced gravity can reveal important insight into locomotion biomechanics. The purpose of this study was to quantify the effects of simulated reduced gravity on muscle activation levels and lower limb biomechanics across a range of overground walking speeds. Our overall hypothesis was that muscle activation amplitudes would not decrease proportionally to gravity level. We recruited 12 participants (6 female, 6 male) to walk overground at 1.0, 0.76, 0.55, and 0.31 G for four speeds: 0.4, 0.8, 1.2, and 1.6 ms$^{-1}$. We found that peak ground reaction forces, peak knee extension moment in early stance, peak hip flexion moment, and peak ankle extension moment all decreased substantially with reduced gravity. The peak knee extension moment at late stance/early swing did not change with gravity. The effect of gravity on muscle activity amplitude varied considerably with muscle and speed, often varying nonlinearly with gravity level. Quadriceps (rectus femoris, vastus lateralis, & vastus medialis) and medial gastrocnemius activity decreased in stance phase with reduced gravity. Soleus and lateral gastrocnemius activity had no statistical differences with gravity level. Tibialis anterior and biceps femoris increased with simulated reduced gravity in swing and stance phase, respectively. The uncoupled relationship between simulated gravity level and muscle activity have important implications for understanding biomechanical muscle functions during human walking and for the use of bodyweight support for gait rehabilitation after injury.

## Introduction

Artificially reduced gravity can reveal principles governing the mechanics and control of legged locomotion in healthy individuals and can enhance rehabilitation in humans and other animals with neurological deficits. Legged locomotion requires the weight of the body to be supported while propulsive forces transferred from the feet to the ground progress the body

Overground_walking_with_simulated_reduced_
gravity/5430234).

**Funding:** The research was partially funded by
National Institutes of Health (https://www.nih.gov/)
R01NS104772 (DPF). The funders had no role in
study design, data collection and analysis, decision
to publish, or preparation of the manuscript.

**Competing interests:** The authors have declared
that no competing interests exist.

forward [1–3]. A reduction in bodyweight allows people with limited strength and muscle coordination to practice walking, develop muscle strength, improve coordination, and build bone density [4,5]. As a result, bodyweight support with a body harness is regularly used in rehabilitation clinics for patients with post-stroke hemiparesis, spinal cord injury, or other neurological disabilities [6–11]. Providing artificial bodyweight support reduces the force, energy, and work demands associated with walking and has been used to study how humans would walk in reduced gravity [12,13]. Prior research with healthy humans has found simulated reduced gravity modifies the walk-to-run transition speed, decreases stance time, and decreases the metabolic cost of walking [1,14–19]. These studies have relied on bodyweight support from a torso harness, and thus, do not affect the gravity level affecting the swing limbs. However, they have proven valuable for identifying biomechanical principles governing legged locomotion.

There are many factors influencing interpretation of past research on the mechanics of simulated reduced gravity or bodyweight supported locomotion. Most studies have focused on treadmill gait, introducing the possibility of substantial horizontal forces being transmitted from the harness to the person. Horizontal forces from a fixed support cable to the harness can alter the function of the muscles and joints independent of the vertical bodyweight support. It is becoming more common to use overground bodyweight support for rehabilitation as it allows the patient the freedom to start, stop, choose their own walking speed, and transition between speeds. Overground simulated reduced gravity also more easily enables practice maneuvering around or over obstacles, going up and down stairs, or transitioning between different types of terrain. Studies that have examined the biomechanics of overground walking with reduced gravity have not exceeded more than 50% bodyweight support [20–22]. This limits the effect on the biomechanical parameters to a smaller range than would be expected at Mars gravity level (38% G). Another factor potentially affecting the outcomes of the studies is the method of bodyweight support. Many rehabilitation systems of bodyweight support do not maintain a constant upward force on the harness, sometimes fluctuating up to 50% or more of the desired bodyweight support level [5,23,24]. Another important factor affecting prior results is the locomotion speed. Walking speed affects muscle activity patterns and amplitudes [25–27]. It is likely that there is a nonlinear relationship between bodyweight support level and walking speed as they affect walking dynamics [15]. Few of the prior studies on walking biomechanics in simulated reduced gravity have studied a range of walking speeds.

Given the potential confounding factors discussed above, it is not surprising that past literature on reduced gravity (or bodyweight supported) human locomotion does not provide many consensus conclusions about limb biomechanics or muscle activity. Most studies do find significant changes in kinematics and kinetics if examining a wide enough range of simulated gravity levels. Trends show reductions in peak ground reaction forces, peak joint moments, and metabolic cost with increasing bodyweight support (or decreasing simulated gravity level) [28–31]. Most studies have found that bodyweight support reduces the activity of some leg muscles, like the medial or lateral gastrocnemius [18,32,33]. Simulations and experimental studies both suggest that these muscles contribute to the support of bodyweight during stance [1,34]. However, some studies have found that muscle activity is independent of bodyweight support level for some muscles. Researchers have suggested that these muscles contribute more to the forward progression of the body and are thus dependent on bodymass and not bodyweight [1]. A few studies have even found that some leg muscles increase electrical activity with bodyweight support. The explanation behind increasing leg muscle activity with reduced bodyweight is unclear, but may be related to co-activation strategies, or changes in kinematic patterns. The discrepancies between studies could be due to different walking speeds, fluctuations in the bodyweight support force, or different choices in measured muscles. Examining

overground walking with a large range of bodyweight support levels that have low fluctuations during the gait cycle would provide a much clearer indication of how ground reaction forces, joint dynamics, and muscle activity, are dependent on bodyweight.

The goal of the study was to determine how simulated reduced gravity influences ground reaction forces, joint dynamics, and lower limb muscle activity when humans walk overground at a range of speeds. We hypothesized that peak vertical and horizontal ground reaction forces, peak joint moments, and peak joint power would decrease with reduced gravity. We also hypothesized that plantarflexor muscle activity would decrease with reduced gravity, and quadriceps muscle activity would not change with gravity level. The rationale for the invariant quadriceps muscle activity hypothesis stemmed from prior observations of simulated reduced gravity treadmill walking and the biomechanical function of the quadriceps to primarily counteract body mass effects and not bodyweight [18,35]. There have been past studies examining muscle activation and/or gait biomechanics walking under simulated reduced gravity, but the range of simulated reduced gravity levels and walking speeds in this study on overground gait make it unique and particularly valuable to our understanding of human locomotion biomechanics.

## Methods

### Data collection

To understand the biomechanics of overground walking with simulated reduced gravity at different speeds, we recruited 12 young, able-bodied participants (6 female, 6 male; 26±4 years of age, body mass 70±8 kg, mean±s.d.). Before participating, each subject signed an informed consent form approved by the University of Florida Institutional Review Board. Participants also signed an optional photo and video release form which was approved by the University of Florida institutional review board. The individuals depicted in the figures and videos have given written informed consent (as outlined in PLOS consent form) to publish these case details. We asked participants to walk at 4 levels of simulated gravity; 1, 0.76, 0.45, 0.31 gravity (G). At each simulated gravity level, participants walked at 0.4, 0.8, 1.2, and 1.6 ms$^{-1}$ over an 8 m long walkway with 3 embedded forceplates (40 cm x 60 cm surface, AMTI, Watertown, MA, USA). Subjects were instructed to walk without looking down at their feet and targeting the forceplates. We did not impart specific instructions on how to walk in simulated reduced gravity to the participants, but we did ask participants to attempt not to include a flight phase in conditions with high levels of reduced gravity and fast walking speeds.

We collected motion capture, ground reaction force, and electromyography data in this study. We recorded forceplate data at 1000 Hz. To measure kinematic data at 100 Hz we used an optical motion capture system (Optitrack, Corvallis, OR, USA) and placed reflective markers on anatomical landmarks (Ilium anterior superior, ilium posterior superior, greater trochanter of the femur, femur lateral epicondyle, femur medial epicondyle, fibula apex of the lateral malleolus, tibia apex of medial malleolus, the posterior of the calcaneus, head of the 1$^{st}$ metatarsal, head of the 5$^{th}$ metatarsal, and hallux) and 4-marker rigid bodies on the thighs and shanks. Wireless surface electromyography (EMG) sensors (Cometa, Bareggio, MI, Italy) measured muscle activation of the tibialis anterior, medial and lateral gastrocnemius, soleus, rectus femoris, vastus lateralis and medialis, and biceps femoris at 1000 Hz. Each sensor had 2 electrodes at a fixed position relative to each other. We placed the sensors on the belly of the appropriate muscle, except for the soleus which was placed below the belly of the muscle to minimize crosstalk with the gastrocnemius. We prepared the skin for electrode placement by shaving the body hair and scrubbing the skin with an alcohol wipe. To minimize movement

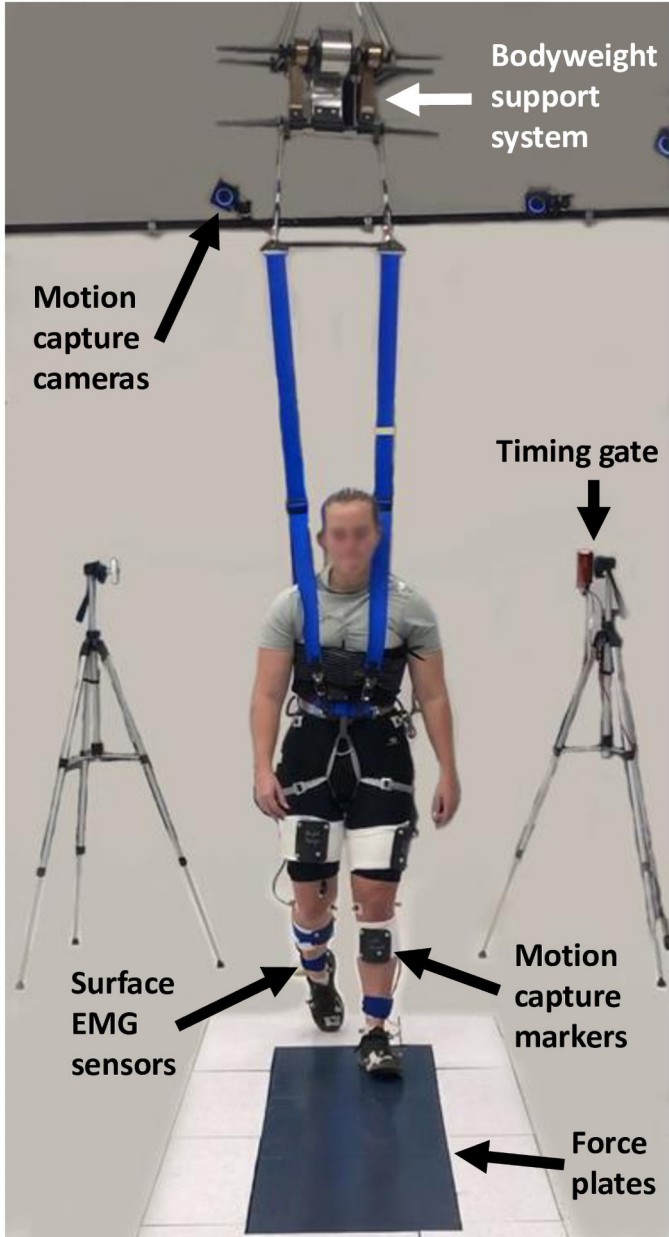

**Fig 1. Experimental set-up.** The particpant walked over 3 overground forceplates. The participant wore 8 surface electromyography (EMG) sensors on each leg. We used motion capture markers to determine kinematics. The participant walked through infrared timing gates before and after the force plates. We used the timing gates to measure walking speed. The bodyweight support system transferred support force to the participant via a modified rock climbing harness.

artifacts, we secured the electrodes by wrapping the leg with athletic wrap. Fig 1 shows the experimental set-up.

To simulate reduced gravity during overground walking we used a low-cost bodyweight support system, described in detail in [36]. Briefly, we suspended constant force springs from a rolling trolley on the ceiling and connected them to the participant via a modified rock-climbing harness. A research assistant ensured the trolley stayed above the user's head as the subject

walked. The constant force springs coiled or uncoiled to accommodate vertical movement of the trunk during gait. Small fluctuations in force were present due to friction, but were less than ±5% of bodyweight. We used a variety of different springs to provide different forces. This method of simulating reduced gravity only affects the overall bodyweight via the harness connection to the torso. It does not alter the gravity level affecting the swinging arms and legs, which is an important difference between our simulated reduced gravity and real reduced gravity [14].

Conditions of speed and gravity level were pseudo-randomized. Participants completed all speed levels at one level of gravity, with 3 minutes of walking to adapt to simulated reduced gravity before data were recorded. During the adaptation period, participants walked back and forth over the walkway. The participant rested for 5–15 minutes before walking at a new gravity level. We defined walking to the subjects as bipedal gait without a flight phase and asked that participants not include a flight phase in any of the trials. We randomized the order of simulated gravity levels and the order of speeds at each gravity level. To ensure the participant walked at the specified speed, we used timing gates to measure their average speed over the forceplates. Good trials were classified as trials wherein the participant struck separate forceplates with their right, then left foot, and their speed was within ±5% of the target speed. We recorded 4 good trials per condition.

Data supporting the findings of this study are openly available at http://figshare.com/collections/Overground_walking_with_simulated_reduced_gravity/5430234.

## Data processing

We defined a stride as right initial contact (heel strike) to consecutive right initial contact. Review of the motion capture data confirmed that initial contact was made by the heel in all trials. Using a cut-off of 18 N, we identified heel strikes from vertical ground reaction force data. In trials where the consecutive right heel strike occurred past the force plates, we used Visual3D to identify second right heel strike. To account for discrepancy between kinematic prediction of initial contact and force plate ground truth in non-typical gait (walking with reduced gravity), we found the delay between the first right heel strike determined by motion capture and ground reaction force. We calculated the timing of the second right heel strike by adding the delay to the consecutive right heel strike found by kinematics. To validate this method, we compared the timing of our delay-offset kinematic predictions with the heel strikes found from vertical ground reaction force and found the discrepancy was less than one frame of motion capture data (0.01 seconds). We extracted right stride length data from Visual3D and normalized it to leg length.

We used Visual3D (C-Motion, Germantown, MD, USA) and MATLAB (Mathworks, Natick, MA, USA) to process the kinematic and kinetic data. Our anthropometric model in Visual3D used default settings to define the segments and included the Coda pelvis model, which estimated the hip joint location [37]. The local coordinate system for each segment was orientated with the Z-axis along the distal to proximal axis, the Y-axis along the posterior-anterior axis, and X-axis directed to the right (orientated medial-lateral on the right leg and vice-versa for the left leg). To calculate ankle angle, we used a separate kinematic model of the foot with a local coordinate system defined by the shank. Ground reaction forces and motion capture data were lowpass filtered at 6 Hz with a 4th order Butterworth filter. Using Visual3D, we calculated joint angles, net internal moments, and powers during the gait cycle. Within this paper, the terminology "joint moment" refers to net internal moment. In MATLAB we extracted peak joint angles, moments and powers and then averaged across trials for each condition. We also extracted peak vertical, medial-lateral, and anterior-posterior ground reaction

force from the force plate data filtered with a lowpass 4[th] order Butterworth filter of 20 Hz. We also normalized all time-series data to 0–100% of stride and averaged for each condition. To prepare the data for comparison across participants, we normalized the force data to body-weight at 1 G and moments & powers to body mass.

We filtered muscle activity data with a zero-lag, 30 Hz highpass, 4[th] order Butterworth filter and then rectified it. For statistical analysis, we calculated and averaged the root mean square (RMS) muscle activity for each participant and condition in stance and swing phase. The RMS muscle activity was normalized with respect to the maximum RMS across conditions (for stance and swing phase independently) and then averaged across participants. We also normalized the time-series EMG data to 1001 data points, representing 0–100% of stride and then averaged across trials for each condition. For visualization purposes, we calculated the moving mean of the data with a window size of 20. We used the moving average window instead of a low-pass filter, as sampling frequency was lost when the data were normalized to 0–100% of stride. For each participant, we normalized the time-series EMG data to their maximum EMG signal across all conditions and then we averaged the data across participants.

We created right-leg sagittal plane force vectors at 3 points during stance-phase using ground reaction force data. The 3 points were found using the vertical ground reaction force and were: the first peak, the minimum after the first peak, and the second peak. We made the sagittal plane vectors by combining the vertical and anterior-posterior ground reaction force at the points of interest.

### Statistical analysis

We used a linear mixed model to test for statistical significance in the biomechanical parameters. We considered gravity level and speed as repeated factors, and participants as a random factor. To adjust for multiple comparisons (significance value of 0.05) on the effect of simulated gravity level we used the Benjamini-Hochberg procedure [38,39].

## Results

Ten of the twelve participants completed all walking speeds at all levels of gravity. Two of the twelve participants were unable to walk at 1.6 ms[-1] for the 0.31 G condition without an aerial phase. As a result, we did not collect data for those two subjects in that condition. Due to temporary equipment failure, some data was missing for select conditions and participants (see S2 Table). We used a subset of 8 participants to confirm the bodyweight support device provided a nearly constant upward force on the humans (within +/- 5% bodyweight) (details in [36]).

### Kinetics

The majority of the kinetic parameters decreased in magnitude with gravity. Reducing the artificial gravity level significantly decreased the maximum vertical and horizontal ground reaction force peaks (all p<0.001) (Table 1 and Fig 2). The sagittal plane ground reaction force vectors changed with bodyweight support (Fig 3). The magnitude of the vectors decreased with reduced gravity, but the vector angles were similar across gravity levels. Figs 4–6 show the averaged joint dynamics data during a stride. The peak hip extension moment, knee flexion moment in early stance, and ankle plantar-flexion moment all declined with gravity (each p<0.001) (Fig 7). Gravity level did not significantly affect peak knee flexion moment during late-stance/early-swing (adjusted p = 0.869). There were significant declines with reduced gravity in peak power generation of the hip, knee in early stance, and ankle, and in peak power absorption of the knee in late stance/early swing (each p<0.001). Table 2 presents the statistical results for the joint kinetic data.

**Table 1. Peak ground reaction force descriptive data from overground walking with bodyweight support.**

| Speed | Simulated gravity level | Peak vertical ground reaction force | | Peak braking force | | Peak accelerating force | | Peak absolute medial-lateral force | |
|---|---|---|---|---|---|---|---|---|---|
| | | Mean | Std.Dev | Mean | Std.Dev | Mean | Std.Dev | Mean | Std.Dev |
| 0.4 ms⁻¹ | 1 G | 1.063 | 0.050 | -0.076 | 0.082 | 0.015 | 0.014 | 0.038 | 0.008 |
| | 0.76 G | 0.842 | 0.049 | -0.063 | 0.062 | 0.015 | 0.015 | 0.031 | 0.009 |
| | 0.45 G | 0.631 | 0.073 | -0.050 | 0.048 | 0.012 | 0.028 | 0.026 | 0.006 |
| | 0.31 G | 0.382 | 0.061 | -0.025 | 0.028 | 0.006 | 0.015 | 0.025 | 0.007 |
| 0.8 ms⁻¹ | 1 G | 1.063 | 0.052 | -0.126 | 0.134 | 0.018 | 0.023 | 0.056 | 0.012 |
| | 0.76 G | 0.853 | 0.063 | -0.100 | 0.108 | 0.019 | 0.025 | 0.051 | 0.010 |
| | 0.45 G | 0.686 | 0.086 | -0.078 | 0.078 | 0.019 | 0.022 | 0.044 | 0.015 |
| | 0.31 G | 0.479 | 0.075 | -0.043 | 0.040 | 0.013 | 0.021 | 0.036 | 0.012 |
| 1.2 ms⁻¹ | 1 G | 1.136 | 0.075 | -0.183 | 0.206 | 0.033 | 0.028 | 0.072 | 0.019 |
| | 0.76 G | 0.990 | 0.078 | -0.155 | 0.179 | 0.030 | 0.030 | 0.062 | 0.013 |
| | 0.45 G | 0.804 | 0.096 | -0.110 | 0.119 | 0.028 | 0.028 | 0.055 | 0.021 |
| | 0.31 G | 0.595 | 0.101 | -0.061 | 0.062 | 0.019 | 0.024 | 0.045 | 0.018 |
| 1.6 ms⁻¹ | 1 G | 1.268 | 0.068 | -0.232 | 0.291 | 0.032 | 0.037 | 0.075 | 0.020 |
| | 0.76 G | 1.141 | 0.103 | -0.212 | 0.254 | 0.039 | 0.049 | 0.068 | 0.019 |
| | 0.45 G | 0.894 | 0.081 | -0.141 | 0.164 | 0.028 | 0.040 | 0.059 | 0.024 |
| | 0.31 G | 0.669 | 0.172 | -0.073 | 0.083 | 0.016 | 0.036 | 0.044 | 0.018 |

Units are bodyweights (BW).

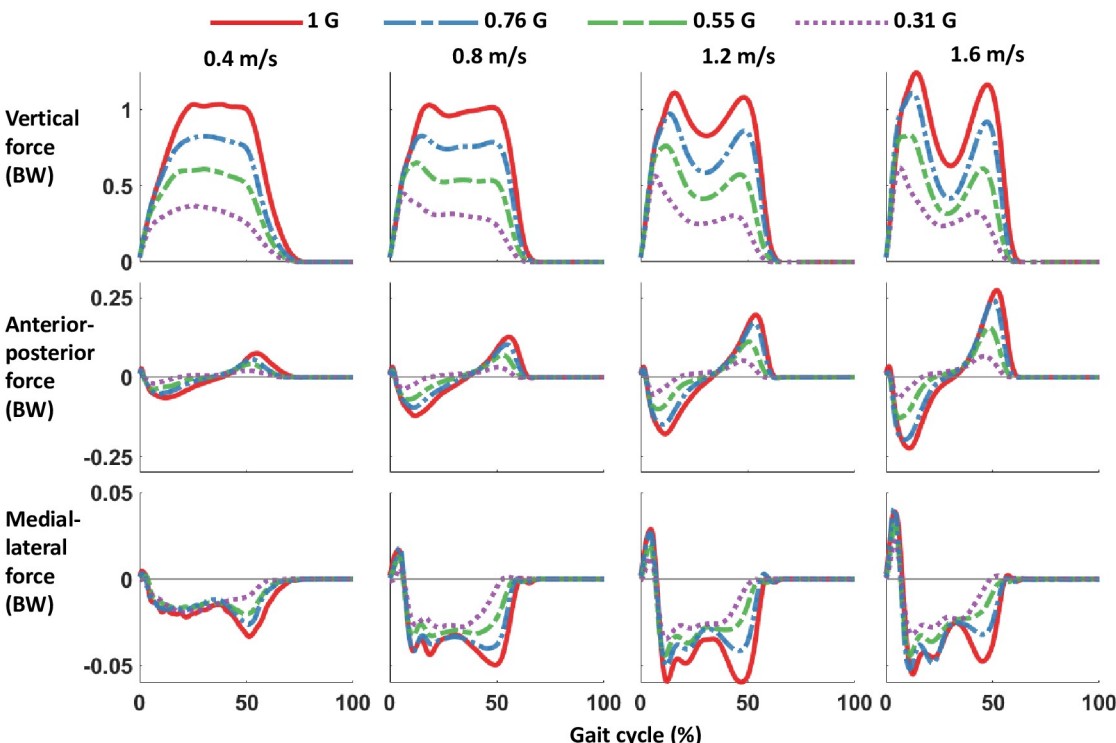

**Fig 2. Average ground reaction forces of 12 participants walking overground with bodyweight support.** Gait cycle was defined as starting with right heel strike (0%) and ending at the next right heel strike (100%). Forces were normalized to bodyweight at 1 G.

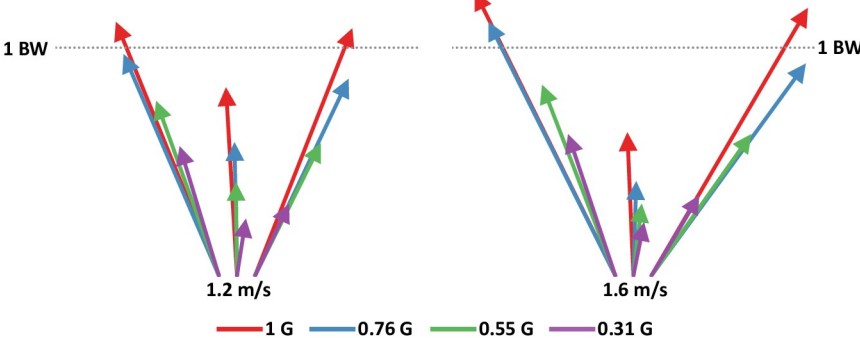

**Fig 3. Saggital plane ground reaction force vectors from walking at 1.2 and 1.6 ms⁻¹ overground with bodyweight support.** The timings of the vectors are: Leftmost vectors found at initial peak of vertical ground reaction force, middle vectors found at the lowest vertical ground reaction force after initial peak, and the rightmost vetors were found at the second peak of vertical ground reaction force. The vectors make the typical 'm' shape associated with the vertical ground reaction force pattern of healthy walking. The dotted horizontal line shows vertical ground reaction force equal to bodyweight. The scale of both plots is equal.

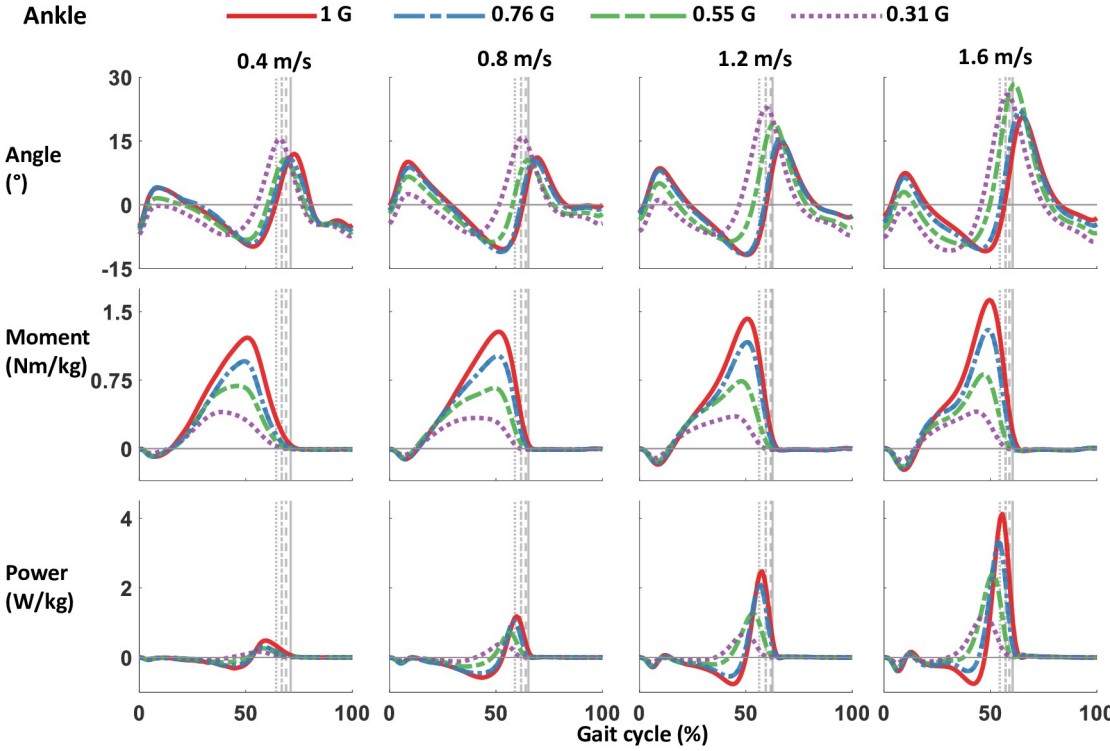

**Fig 4. Ankle joint angle, moment, and power from walking with bodyweight support at 4 speeds.** Data were averaged from 12 participants, and gait cycle was right heel strike to consecutive right heel strike. The grey lines represent right toe off at different levels of simulated gravity. Bodyweight support is shown as gravity, with normal walking at 1 G, and bodyweight support of 69% bodyweight as 0.31 G. Positive angle and moment is plantarflexion, negative angle and moment is dorsiflexion. The ankle angle in neutral standing position is 0˚. Positive power is power generated, negative power is power absorbed. Magnitude of moment and power decrease with simulated gravity.

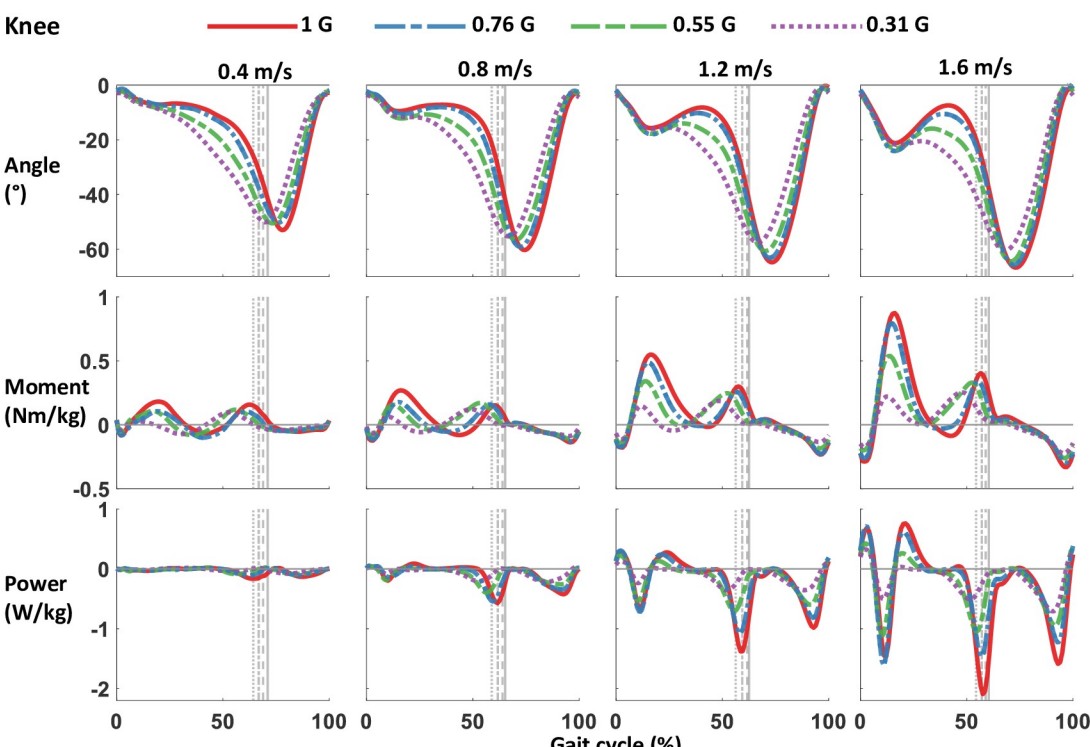

**Fig 5. Knee joint angle, moment, and power from walking with bodyweight support at 4 speeds.** Data were averaged from 12 participants, and gait cycle was right heel strike to consecutive right heel strike. The grey lines represent right toe off at different levels of simulated gravity. Bodyweight support is shown as gravity, with normal walking at 1 G, and bodyweight support of 69% bodyweight as 0.31 G. Positive angle and moment is knee extension, negative angle and moment is knee flexion. Positive power is power generated, negative power is power absorbed. Magnitude of moment and power tends to decrease with simulated gravity.

## Muscle activity

The effect of simulated reduced gravity on muscle activity was less straightforward than the gait kinetics. Figs 8 and 9 show that some muscles decreased activity amplitude in response to reduced gravity, while other muscles had increased activity amplitude. Some of the muscles did not change their activity amplitude with gravity. Table 2 presents the EMG statistical results by muscle. We found reduced gravity significantly decreased medial gastrocnemius, rectus femoris, vastus medialis, and vastus lateralis activity during stance phase ($p < 0.001$). We found that decreased gravity significantly increased biceps femoris activation in stance phase and tibialis anterior activation in swing phase ($p = 0.050$ and $p < 0.001$, respectively).

## Stride length

Stride length did not change with gravity level ($p > 0.99$) (Fig 10 & Table 2). Stride length increased with walking speed.

## Discussion

Reducing gravity induced large decreases in ground reaction force and net muscle moment peaks during human walking, but there was not a concomitant and consistent decrease in muscle activation amplitude. Results supported our first hypothesis that ground reaction forces, joint moments, and joint power would decrease with gravity. Providing artificial

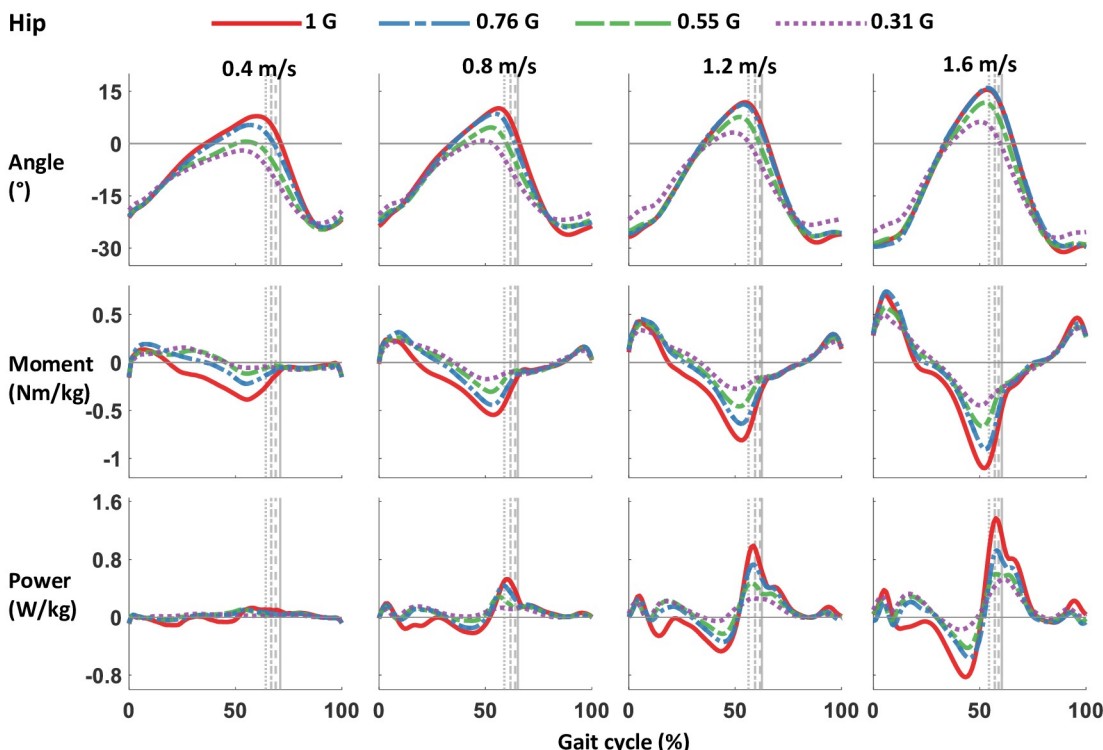

**Fig 6. Hip joint angle, moment, and power from walking with bodyweight support at 4 speeds.** Data were averaged from 12 participants, and gait cycle was right heel strike to consecutive right heel strike. The grey lines represent right toe off at different levels of simulated gravity. Bodyweight support is shown as gravity, with normal walking at 1 G, and bodyweight support of 69% bodyweight as 0.31 G. Positive angle and moment is hip extension, negative angle and moment is hip flexion. Hip angle in neutral standing position is 0˚. Positive power is power generated, negative power is power absorbed. Magnitude of moment and power decrease with simulated gravity.

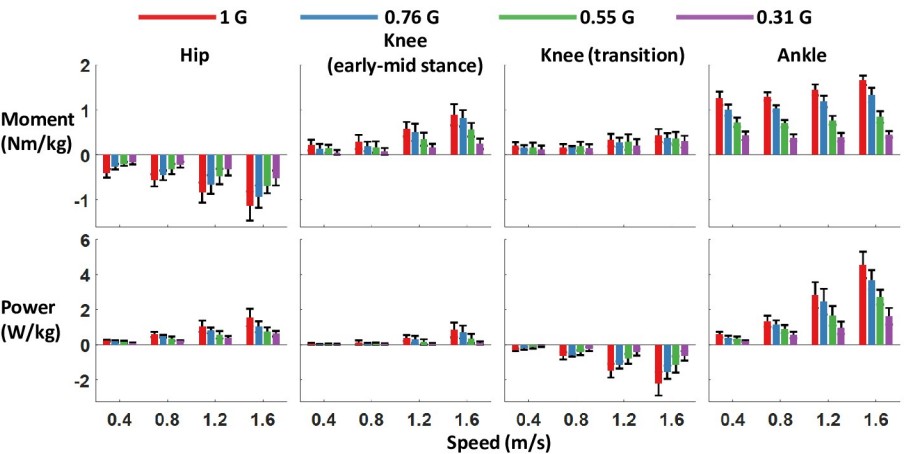

**Fig 7. Peak joint moment and powers from walking overground with bodyweight support.** Error bars are 1 standard deviation. Early-mid stance knee extension moment is the initial peak of knee extension moment. Transition knee extension moment is the peak extension moment that occurs around push-off (the second peak). Bodyweight support had a signigicant effect on hip, early-mid stance knee, and ankle moments (p < 0.01). All peak joint powers were also reduced with decreasing gravity (p < 0.001).

**Table 2. Statistical results of significance of gravity level on dependent variables.**

| | Variable | Adjusted p value |
|---|---|---|
| Force | Maximum vertical | <0.001 * |
| | Maximum braking | <0.001 * |
| | Maximum accelerating | <0.001 * |
| | Maximum absolute medial-lateral | <0.001 * |
| Moment | Hip flexion | <0.001 * |
| | Knee extension (early-mid stance) | <0.001 * |
| | Knee extension (transition) | 0.869 |
| | Ankle plantarflexion | <0.001 * |
| Power | Hip generation | <0.001 * |
| | Knee generation (early-mid stance) | <0.001 * |
| | Knee absorption (transition) | <0.001 * |
| | Ankle generation | <0.001 * |
| Stance phase RMS EMG | Rectus femoris | <0.001 * |
| | Vastus lateralis | 0.009 |
| | Vastus medialis | 0.007 |
| | Biceps femoris | 0.050 |
| | Lateral gastrocnemius | >0.99 |
| | Medial gastrocnemius | <0.001 * |
| | Soleus | >0.99 |
| | Tibialis anterior | 0.762 |
| Swing phase RMS EMG | Rectus femoris | >0.99 |
| | Vastus lateralis | >0.99 |
| | Vastus medialis | >0.99 |
| | Biceps femoris | >0.99 |
| | Tibialis anterior | <0.001 * |
| Spatial | Right stride length | >0.99 |

The Benjamini-Hochberg procedure adjusted the p values for multiple comparisons. p values less than 0.05 were considered significant.

* indicates that the effect of gravity was significant.

reduced gravity decreased bodyweight of the participants, requiring them to generate less force against the ground to support and vertically accelerate their bodies. We had also hypothesized that plantarflexor muscle activity would decrease with gravity as past research had suggested the plantarflexor muscles play a primary role in supporting bodyweight [1,18]. Medial gastrocnemius EMG amplitude was smaller in the stance phase at lower gravity levels, but the lateral gastrocnemius and soleus EMG amplitude did not change with gravity. This suggests that the division of mechanical function of the plantarflexors may be more nuanced than previously assumed. Based on a prior study on humans walking on a treadmill with reduced gravity, we had hypothesized that the quadriceps muscles would not show decreases in EMG amplitude with gravity [18]. In contrast to the hypothesis, we found that quadriceps EMG amplitude decreased with gravity. This finding suggests that the vastii muscles aid with supporting bodyweight during stance in addition to helping to accelerate body mass.

We found no significant effect of simulated gravity level on stride length, but stride length did increase with walking speed. Donelan and Kram [15] also found that the influence of walking speed on stride length was stronger than that of gravity level. Their results indicated a slight trend of decreased stride length with decreasing gravity level, which is not reflected in our

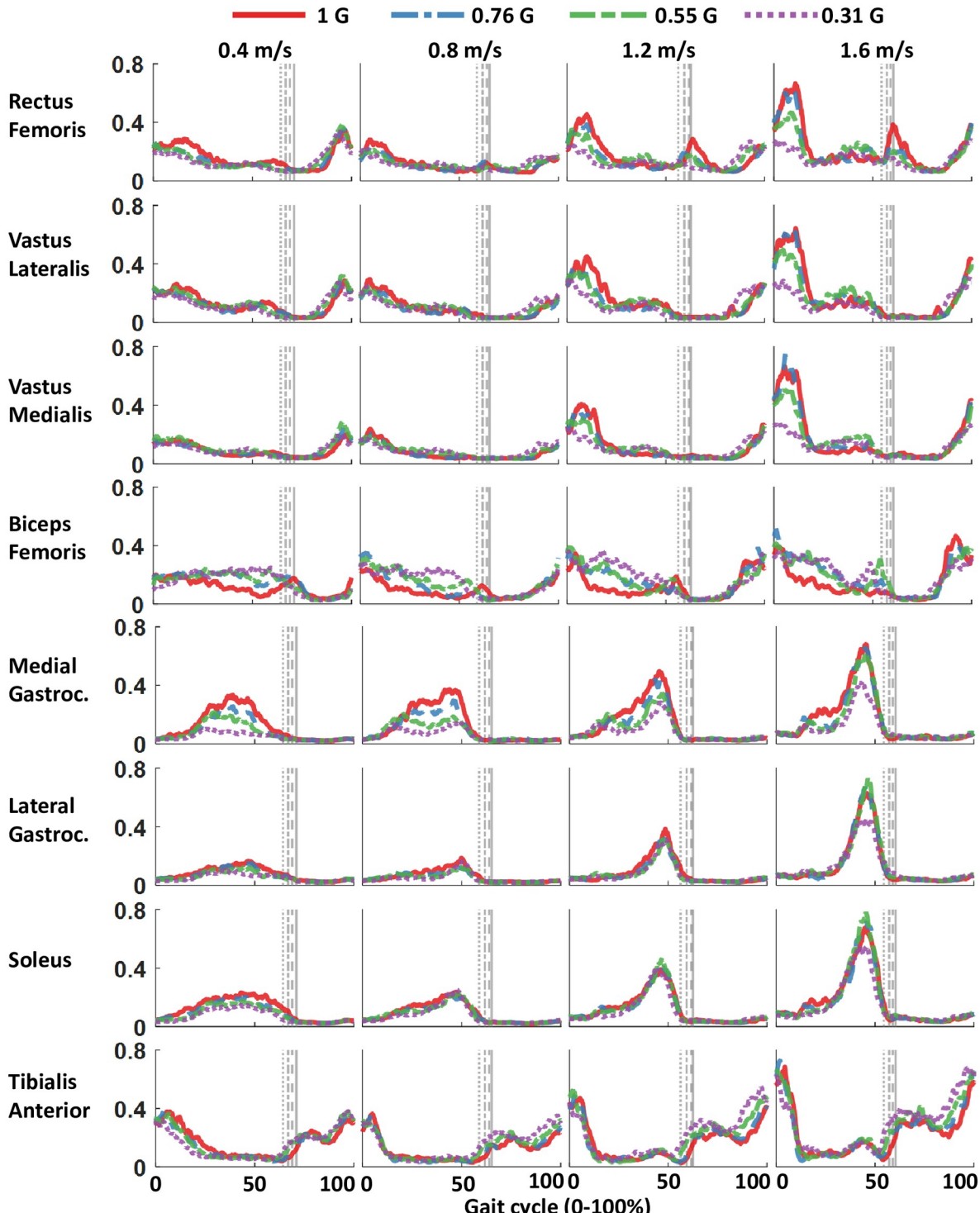

**Fig 8. Time series muscle activation in response to bodyweight support for overground walking at 4 speeds.** The linear envelopes are data averaged from 12 participants. Muscle activation was normalized at the individual participant level to maximum activation across all walking conditions. Gait cycle was right heel strike to consecutive right heel strike. The grey lines represent right toe off at different levels of simulated gravity All plots have the same axes.

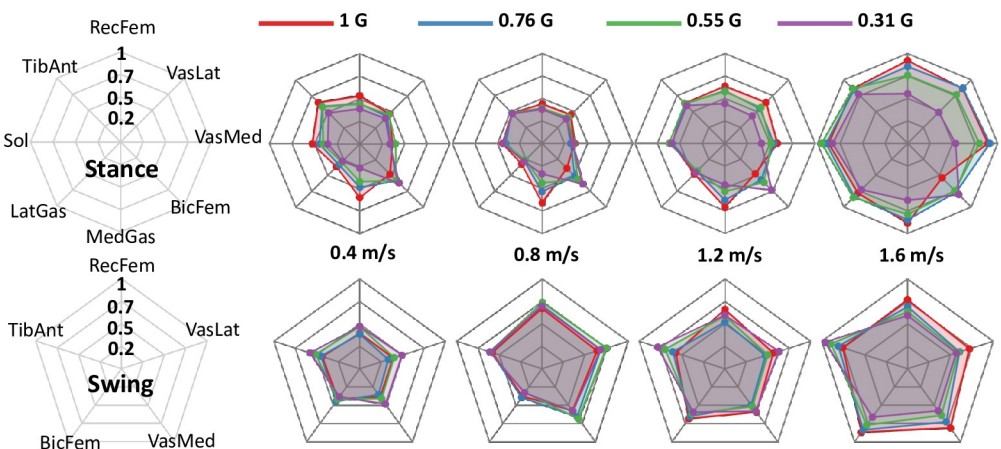

**Fig 9. Response of RMS muscle activity to bodyweight support at 4 different overground walking speeds.** Data is presented as a spider plot. Each spoke on the compass represents a muscle. The circular bands represent level of normalized activation, with the smallest activation at the innermost band and a maximum activation of 1 at the outer-most band. RMS data was normalized to maximum RMS activation of each muscle at the participant level. The top row represent the RMS data in stance phase, and the bottom row of spider plots represent the data in swing phase. The shortans for the muscles are: Rectus femoris (RecFem), Vastus lateralis (VasLat), Vastus medialis (VasMed), Biceps femoris (BicFem), Medial gastrocnemius (MedGas), Lateral gastrocnemius (LatGas), Soleus (Sol), and Tibialis anterior (TibAnt).

data. One reason for the discrepancy may be that we had higher stride length variability due to the less controlled nature of walking overground, whereas Donelan and Kram [15] used a treadmill with precisely controlled speed.

Reduced gravity significantly decreased maximum vertical, braking, accelerating, and medial-lateral ground reaction forces across walking speeds. The artificial reduced gravity effectively reduced the bodyweight of the participant, leading to lower forces between the foot and ground in all three axes. These findings are in agreement with prior simulated reduced gravity studies on human walking. The benefit of smaller ground reaction force magnitude is reduced lower limb joint loading during walking. Decreased load-bearing can reduce pain from walking with arthritis [40,41], facilitate walking in patients who cannot generate large enough muscle forces to walk in normal gravity [5,42–44], or allow a person with an ampu-tated leg to acclimatize to walking with a prosthetic with less pain at the human-prosthesis

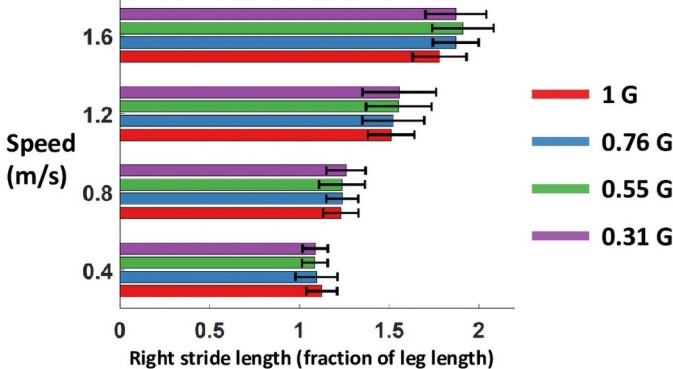

**Fig 10. Average and standard deviation of right stride length across conditions.** The stride length is normalized to leg length.

interface [45–47]. The sagittal plane vectors (Fig 3) show that the orientation of the ground reaction force stayed similar across bodyweight support levels. This is in keeping with prior research showing that humans maintain a consistent ground reaction force vector direction during running in reduced gravity [48].

The reduction of ground reaction forces with reduced gravity resulted in significantly smaller sagittal plane net joint moments and joint powers during most of the stance phase. Gravity level decreased knee extension moment in early-mid stance, ankle plantarflexion moment, and hip flexion moment. We also found reduced mechanical power absorption and generation in the lower limb joints. Smaller joint powers suggest that less muscle work was performed at reduced gravity compared to normal gravity, assuming no increase in the amount of muscular co-contraction. We found no evidence of increased co-contraction in our muscle activity results. Only the biceps femoris in stance phase and the tibialis anterior during swing phase had increased activity with decreasing gravity.

The peak knee extension moment around the transition to swing phase did not change with gravity. At the fastest walking speeds, there was a noticeable decrease in rectus femoris (knee extensor) activity at the beginning of swing phase with reduced simulated gravity level, but gravity did not change rectus femoris activity at the slower speeds, and no overall significance of gravity on RMS rectus femoris (or vastii) activity in swing phase was found. Winter [49] reported that this knee extension moment prevents undesirable knee flexion during swing initiation, allowing the trailing limb to redirect the center of mass at the stance-swing transition [50]. By opposing flexion at the knee, the contraction of the plantarflexors is more effectively directed to increase ankle plantarflexion and prevent energy loss at the knee joint. Our results support this interpretation. Underwater walking experiments corroborate our findings as the knee moment only peaked in extension in late stance to assist with forward propulsion [51,52].

Gravity level affected muscle activation differently across all the muscles (Fig 9 & Table 2). We found that reducing gravity decreased the EMG activity of three knee extensor muscles and one plantarflexor muscle during stance phase. Two of the plantarflexor muscles were not statistically different with gravity level. Reducing gravity actually increased EMG activity of the dorsiflexor during the swing phase and EMG activity of a knee flexor during the stance phase. Overall, three of the eight muscles did not show significant differences in EMG amplitude during stance with respect to gravity level, and seven of the eight muscles did not show significant differences in EMG amplitude during swing with respect to gravity level.

Contrary to our initial hypothesis, we found quadriceps muscle activity decreased during the stance phase with reduced gravity. The reduction in rectus femoris, vastus lateralis, and vastus medialis occurred during early stance (see Fig 8) and was more pronounced at faster walking speeds. The reduction in activation occurred near the initial peak knee extension moment, which significantly decreased with reduced gravity. Past studies have not provided a consensus on the effect of bodyweight support on quadriceps activation (Table 3). It seems that the differences across studies in quadriceps muscle activation with bodyweight support are likely due to differences in walking speed and bodyweight support mechanism. Our results indicate that decreases in quadriceps activation with reduced gravity were more prominent at faster speeds (Figs 8 and 9), which seems to concur with the previous studies. Studies with similar walking speeds report dissimilar findings of the quadriceps response to gravity (ex. [53] found reducing gravity increased quadriceps activation at 0.56 m/s and 0.83 m/s, while [55] found decreased activation at 0.7 ms). The method of providing bodyweight support or simulating reduced gravity is therefore likely to impact the effect of bodyweight support level on muscle activation.

**Table 3. Effects of bodyweight support on muscle activity found in previous studies.**

| Study | Walking Method | Bodyweight Support System | Bodyweight support levels | Walking Speed (m/s) | Muscle | Result with increased bodyweight support (reduced gravity level) |
|---|---|---|---|---|---|---|
| McGowan et al. [1] | TM | Low-stiffness rubber tubing springs | 0, 25% BW | 1.3 | Medial gastroc. | Decreased in support phase |
| | | | | | Soleus | No change |
| Ferris et al. [18] | TM | Low-stiffness rubber tubing springs | 1, 0.75, 0.5, 0.25 G | 1.25 | Vastus lateralis | No change |
| | | | | | Soleus | Decreased |
| | | | | | Medial gastroc. | Decreased |
| | | | | | Tibialis anterior | No change |
| Colby et al. [19] | TM | Unspecified | 0, 20, 40% BW | 1.34 | Quadriceps | Decreased (stance) |
| | | | | | Gastrocnemius | No change (stance) |
| | | | | | Hamstrings | No change (stance) |
| Mun et al. [22] | OG | Robotic walker | 0, 10, 20, 30, 40% BW | between 0.37–0.41 | Gastrocnemius | Decreased (stance) |
| | | | | | Soleus | No change (stance) |
| | | | | | Vastus Med. | No change (stance) |
| | | | | | Rectus Femoris | No change (stance) |
| | | | | | Biceps femoris | No change (stance) |
| | | | | | Tibialis anterior | No change (swing) |
| Lewek [30] | TM | Low-stiffness rubber tubing springs | 0, 20, 40% BW | 0.4, 0.6, 0.8, 1, 1.2, 1.4, 1.6 | Medial gastroc. | No change |
| | | | | | Lateral gastroc. | No change |
| | | | | | Soleus | No change |
| Dietz and Columbo [32] | TM | Counterweight | 0, 25, 50, 75% BW | 0.34 | Medial gastroc. | Decreased |
| | | | | | Tibialis anterior | No change |
| Fischer et al. [33] | OG | Biodex Unweighing System | 0, 15, 30% BW | ~1.15 | Lateral gastroc. | Decreased |
| | | | | | Tibialis anterior | Decreased |
| | | | | | Vastus lateralis | Decreased |
| | | | | | Rectus femoris | No change |
| Ivanenko et al. [53] | TM | WARD pneumatic bodyweight support system [54] | 0, 35, 50, 75, 95% BW | 0.19, 0.31, 0.56, 0.83, 1.39 | Lateral gastroc. | Decreased |
| | | | | | Biceps femoris | Increased between 75 and 95% BW. Speed dependent between 0 and 75% BW. |
| | | | | | Vastus lateralis | Speed 0.19 & 0.31: No change<br>Speed 0.56 & 0.83: Increased<br>Speed 1.39: Decreased |
| | | | | | Rectus femoris | Speed 0.19 & 0.31: No change<br>Speed 0.56 & 0.83: Increased<br>Speed 1.39: Decreased |
| | | | | | Tibialis anterior | Speed 0.19 & 0.31: No change<br>Speed 0.56 & 0.83: Increased<br>Speed 1.39: Increased between 0 and 75% BW.<br>Increased between 75 and 95% BW. |
| Kristiansen et al. [55] | TM | Lower body positive pressure | 0, 20, 40, 60, 80% BW | 0.7, 1 | Soleus | Decreased |
| | | | | | Lateral gastroc. | Decreased |
| | | | | | Medial gastroc. | Decreased |
| | | | | | Tibialis anterior | Decreased |
| | | | | | Vastus medialis | Decreased |
| | | | | | Vastus lateralis | Decreased |
| | | | | | Biceps femoris | No change |

TM is treadmill, OG is overground walking, and gastrocnemius has been shortened to gastroc.

We found the response of plantarflexors to gravity to vary between individual muscles. The medial gastrocnemius was the most responsive to gravity level, while soleus and lateral gastrocnemius did not vary with gravity. Our data indicates that the reduction in medial gastrocnemius activation was more pronounced at slower speeds than at faster speeds. Fig 8 suggests that the effect of gravity was nonlinear on the plantarflexor muscles. At the fastest walking speed, the biggest drop in plantarflexor muscle EMG amplitude was between 0.55 G and 0.31 G data, with little differences between 1.0, 0.76, and 0.55 G gravity levels. There have been conflicting findings in the literature on plantarflexor EMG response to gravity (Table 3). Each of these studies appears to tell a slightly different story, and it is unclear why. Although sensitivity to bodyweight support tends to change with speed, there is no consistent response of the plantarflexors to gravity. The method of bodyweight support (counter-weight, dynamic with high force fluctuations, dynamic with low fluctuations, positive pressure) and the magnitude of the bodyweight support likely contribute to changes in kinematics and kinetics that affect the underlying muscle activation. Many studies do not report results for all three plantarflexors, making it difficult to understand if there is an overall effect of gravity on the plantarflexors.

Despite only the medial gastrocnemius showing a significant decrease in EMG activity among the plantarflexor muscles with reduced gravity, ankle plantarflexion moment and power during stance substantially decreased with gravity. A possible explanation for the decrease in plantarflexor moment without an accompanying decrease in plantarflexion muscle activity may be that the plantarflexors were operating at a less effective point of the force-length relationship at reduced gravity levels. Maximum muscle force is dependent on both the muscle length and velocity of the contraction [49,56,57]. If gravity level influenced the length and speed of muscle contraction, then electromyography values may not change even though the resultant muscle force may be different. Recent combinations of ultrasound imaging with musculoskeletal modeling show that the medial gastrocnemius, lateral gastrocnemius, and soleus do not all behave similarly in response to altered mechanical demand at the ankle [58]. Specific reasons for this differential would need to be studied with ultrasound or computer simulations to fully understand how the muscles are changing within the force-length-velocity relationship estimate muscle fascicle length and velocity during walking at different gravity levels [59,60].

We found reducing gravity increased biceps femoris activation during stance phase. The biceps femoris is a biarticular muscle that flexes the knee and extends the hip. Decreased gravity led to a reduced hip flexion moment during stance phase. Increased bodyweight support also decreased knee extension angle in mid-stance and reduced knee flexion moment in early-mid stance. The increase in biceps femoris activation could be a compensation for the reduced activation of the medial gastrocnemius—another muscle that flexes the knee. The majority of studies have found that bodyweight support level does not affect biceps femoris or hamstrings activation (Table 3). An overground walking study at ~0.4 ms$^{-1}$ used a robotic walker to provide bodyweight support up-to 40% bodyweight and found increased bodyweight support had no impact on biceps femoris activation, reduced knee flexion, and increased hip extension [22]. To the best of our knowledge, no studies have found reduced gravity to decrease biceps femoris activation.

Contrary to previous findings, we found the tibialis anterior activation to increase in swing phase with decreasing gravity. We also noted that the ankle tended to stay more dorsiflexed during swing phase when at reduced gravity levels. The consensus of past research is that bodyweight support does not affect tibialis anterior activation in swing phase (Table 3). Our finding of increased tibialis anterior activation in swing phase was less pronounced at slower speeds and seems likely to be related to different physical demands of overground locomotion compared to treadmill locomotion with bodyweight support.

Although it is clear from both our data and the findings of previous studies that walking speed interacts with the effect of simulated gravity on muscle activity, speed alone cannot account for the dissonance in findings between studies. The method of bodyweight support and the walking environment are partially responsible for the variance in findings between studies. Different bodyweight support systems and reduced gravity simulators have distinctive fluctuations in vertical support force and medial-lateral and anterior-posterior pulling forces. The profile of the applied force affects the load carried by the legs and thus alters the demands on the muscles and muscle activity. The bodyweight support systems can also introduce differences in the mass and inertia of the system. Combining a fixed bodyweight support system or reduced gravity simulator with a treadmill can exaggerate the problems with force fluctuations as the person is able to move position on the treadmill, while the position of the support system is fixed. A further consideration of walking environment on the effect of simulated gravity is that fluctuations in treadmill belt speed can transfer energy to the user, thus impacting muscle activity [61,62].

The decreases in net muscle moment with simulated reduced gravity did not occur with proportional changes in muscle activation. Although joint torques greatly decrease, the activity of muscles that contribute to generating the torque about the joint, did not decrease to the same degree. There is a long history of research trying to predict the relationship between net muscle moments and electromyographic activity [63–66], and there are many confounding factors that make the prediction difficult at best and impossible at worst. Recruitment of antagonist muscles, changes in muscle velocity and muscle length, prior activation history, and selective recruitment of different types of muscle fibers are some of the issues that make predictions of net muscle moments from EMG amplitudes problematic. In our experimental situation, many of those factors are likely not major factors as the individuals were performing similar movement tasks involving similar neural control substrates and feedback pathways. Regardless, it is important to note the large variation between muscle activity amplitudes and net muscle moments for walking at a range of simulated gravity levels. Both rehabilitation and human spaceflight studies have often used muscle activity as simple indicators of the mechanical demands placed on muscles. Our results show that this basic assumption is not valid even under very similar locomotor tasks.

When examining the relationship between muscle activation amplitude and joint moments, we calculated only the net internal moment in the sagittal plane. Activation of the leg muscles can also contribute to joint moments in the frontal and transverse plane. For example, the bi-articular biceps femoris flexes the knee and extends the hip, but it also adducts both the hip and knee, externally rotates the hip, and internally rotates the knee [35]. Muscles with increased activation or no response to simulated reduced gravity may have maintained activation amplitude to stabilize joints in the transverse or frontal planes. Future examination of abduction/adduction, and rotational moments in simulated reduced gravity may provide more insight into the roles of muscles in locomotion.

The changes in muscle activation related to reduced gravity suggest that some assumptions about the reflex pathways during human walking are not fully understood. Previous research on cats and humans has implicated afferent sensors in the recruitment of extensor muscles during stance [67–73]. Both 1b (Golgi tendon organs) and 1a afferents (muscle spindles) can respond to increased muscle loading to increase neural recruitment of the stance limb extensor muscles. In our experimental situation, it could be expected that reducing gravity and the ground reaction forces would have led to decreased excitation of the stance limb muscles. Our experiments found that reduced gravity decreased the ground reaction force magnitude and the ankle extension moment, but the activity of 2 of the 3 plantarflexors did not change with gravity. This would suggest that either other excitatory pathways (e.g. supraspinal drive, spinal

neural oscillating circuits) compensated for the reduced excitatory stimulation from the positive force feedback via 1b afferents, or that the response is less linear than previously theorized. It could be a linear feedback pathway with a low-level ceiling so that any loading over a certain threshold produces the same stimulation.

These findings are useful in understanding the implications of overground walking with simulated reduced gravity for rehabilitation. The reduction in ground reaction forces and joint moments are beneficial for patients with fragile bones, joint replacements, or limb prostheses. For patients with muscular weakness or difficulty controlling muscle contraction, bodyweight support can facilitate walking when it is not possible at normal gravity. One of the fundamental principles of neurological rehabilitation is to enhance neuroplasticity by promoting Hebbian learning. This requires motor neurons to fire repeatedly with gait practice, allowing for improved recruitment patterns and coordination. The data from this study indicate that there is not a linear and proportional relationship between gravity level and muscle activation amplitude. Many of the muscles in the lower limb that power walking maintain relatively high activity amplitudes even when bodyweight is reduced to a third of normal via a harness. This suggests the use of the artificial reduced gravity will not drastically restrict recruitment of motor neurons. Future studies would have to confirm similar results in different patient populations, but the results from the neurologically intact subjects in this study are supportive of using high levels of bodyweight support if needed.

We found no fundamental changes to locomotor pattern due to simulated reduced gravity. The patterns of joint angle, moment, power, and muscle activity remained relatively consistent across gravity levels. Furthermore, participants reported no difficulty in learning to walk with reduced gravity and appeared to adapt quickly. The similar locomotor patterns and feedback from participants lead us to believe there was no altered neuromuscular control, which is a promising finding in the context of using overground reduced gravity walking as a rehabilitation tool. The level of simulated gravity should not alter the locomotor task and thus the improvements in neuromuscular control achieved through rehabilitation with reduced gravity would be transferable to walking without bodyweight support.

There were some limitations to this study that could have affected the outcome. Participants had 3 minutes to adapt to walking in simulated reduced gravity before we recorded data. A longer training time may have changed the kinematic and kinetic patterns adopted by the participants. Based on findings by Donelan and Kram [15], who found relatively steady kinematic patterns after 1 minute of walking in reduced gravity, we determined 3 minutes to be a reasonable time for gait adaption to occur. A larger number of participants would have increased the statistical power of our findings. However, the most important findings of our study focused on large magnitude changes in gait biomechanics with concomitant small or no changes in electromyography amplitudes. Increased statistical power could have statistically validated the small changes in electromyography amplitudes but it would not have altered the main conclusions of the study. The number of participants in this study was similar or greater than many previous studies on locomotion biomechanics under simulated reduced gravity [13,15,18,74–76]. A third limitation is that we only examined the muscle activity of 8 superficial lower leg muscles. It would have been beneficial to examine the effect of reduced gravity on more leg muscles, including non-superficial muscles that require indwelling EMG sensors.

Another limitation of our study is that we only used one means for providing the simulated reduced gravity. This made it more difficult to compare our results to other studies that have used other types of bodyweight support systems (e.g. counterweights, pneumatics, or treadmill based systems). A future study could use multiple types of bodyweight support systems, both overground and on a treadmill, to understand how the mechanism of simulating reduced gravity effects locomotion biomechanics and control. A recent study by Barela *et al.*, examined

changes in temporal-spatial parameters between overground and treadmill walking with body-weight support up to 20% bodyweight, but used a different method of bodyweight support for treadmill and overground walking [20]. That approach does not make direct comparison easy given multiple changing factors. Our simulation did not affect the gravity affecting the swing-ing limbs, so was not a true representation of how reduced gravity on Mars or other planets would affect muscle function. Thus, our findings need to be interpreted with that limitation in mind. Lastly, it would be helpful to include ultrasound measurements of muscle fascicle length in the future to better provide indication of muscle function during walking in simulated reduced gravity. Despite the limitations of our study, it did provide some clear outcomes and suggests how future studies could better identify the relationships between muscle function and simulated reduced gravity levels.

## Concluding remarks

Using simulated reduced gravity, we found that there is not a one-to-one correlation between ground reaction forces and lower limb muscle activity amplitudes for humans walking at a range of gravity levels. Across a range of speeds, humans walked with smaller ground reaction forces and joint moments at reduced gravity levels. However, lower limb muscle EMG ampli-tudes did not always show clear reductions with reduced gravity. The magnitude of EMG reductions when they did occur during stance were not linear and proportional to the ground reaction forces or peak joint moments. Some muscles, such as the soleus and the lateral gas-trocnemius, showed no statistically significant differences in EMG amplitude with a three-fold reduction in gravity level. These findings have implications for understanding muscle function during human locomotion and for neurological rehabilitation of human locomotion using bodyweight support harnesses to facilitate overground stepping.

## Supporting information

**S1 Table. The unadjusted p-values of the effects of simulated gravity level (gravity), walk-ing speed (speed), and the interaction between walking speed and gravity level on depen-dent variables.**
(DOCX)

**S2 Table. Missing data in conditions.** Table shows the number of participants for whom data is missing. No number means no data were missing for that condition and data type. The abbreviated headings of the columns are: tibialis anterior (TA), soleus (Sol), medial gastrocne-mius (MGas), lateral gastrocnemius (MGas), rectus femoris (RecF), vastus medialis (VM), vas-tus lateralis (VL), and biceps femoris (BF).
(DOCX)

**S1 File. WalkingWithRedGrav(0.3G)V2.** Supplementary Video- Experimental set up with a participant walking at 0.31 G.
(MP4)

## Author Contributions

**Conceptualization:** Mhairi K. MacLean, Daniel P. Ferris.

**Data curation:** Mhairi K. MacLean.

**Formal analysis:** Mhairi K. MacLean.

**Funding acquisition:** Daniel P. Ferris.

**Investigation:** Mhairi K. MacLean.

**Methodology:** Mhairi K. MacLean, Daniel P. Ferris.

**Project administration:** Mhairi K. MacLean, Daniel P. Ferris.

**Resources:** Daniel P. Ferris.

**Supervision:** Daniel P. Ferris.

**Visualization:** Mhairi K. MacLean.

**Writing – original draft:** Mhairi K. MacLean.

**Writing – review & editing:** Mhairi K. MacLean, Daniel P. Ferris.

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
