## [Decision Letter · Decision Letter 0]

5 Mar 2021

PONE-D-21-02036

Human muscle activity and lower limb biomechanics of overground walking at varying levels of simulated reduced gravity and gait speeds

PLOS ONE

Dear Dr. Ferris,

Thank you for submitting your manuscript to PLOS ONE. After careful consideration, we feel that it has merit but does not fully meet PLOS ONE’s publication criteria as it currently stands. Therefore, we invite you to submit a revised version of the manuscript that addresses the points raised during the review process.

We look forward to receiving your revised manuscript.

Kind regards,

Pei-Chun Kao

Academic Editor

PLOS ONE

Journal Requirements:

2. Please correct your reference to "p=0.000" to "p<0.001" or as similarly appropriate, as p values cannot equal zero.

3.In your Data Availability statement, you have not specified where the minimal data set underlying the results described in your manuscript can be found. PLOS defines a study's minimal data set as the underlying data used to reach the conclusions drawn in the manuscript and any additional data required to replicate the reported study findings in their entirety. All PLOS journals require that the minimal data set be made fully available. For more information about our data policy, please see http://journals.plos.org/plosone/s/data-availability.

4. Please include a caption for figure 10.

Reviewers' comments:

Reviewer's Responses to Questions

**Comments to the Author**

1. Is the manuscript technically sound, and do the data support the conclusions?

Reviewer #1: Yes

Reviewer #2: Yes

2. Has the statistical analysis been performed appropriately and rigorously? 

Reviewer #1: Yes

Reviewer #2: Yes

3. Have the authors made all data underlying the findings in their manuscript fully available?

Reviewer #1: Yes

Reviewer #2: Yes

4. Is the manuscript presented in an intelligible fashion and written in standard English?

Reviewer #1: Yes

Reviewer #2: Yes

5. Review Comments to the Author

Reviewer #1: Dear Author,

This is a well written and comprehensive paper. The rationale and discussion are clear and concise. Thank you.

I have a few minor comments that would help the reader.

General comments:

It would be useful to include a brief description of the marker set, muscles studied, and EMG placement in the methods. As well as a description of the method used to calculate the hip joint centres, the equations used to calculate the anthropometric model, and the coordinate system of segments.

See a recent publication by Derrick et al. (https://doi.org/10.1016/j.jbiomech.2019.109533)

Specific comments:

Line 14: Replace "gait biomechanics" with "lower limb biomechanics".

Line 153: Should it be stride length instead of step length?

Line 158-159: Replace semi-colons with commas.

Line 302-302: Do you mean "The benefit of smaller ground reaction force magnitude is reduced lower limb joint loading during walking?"

Using lower limb instead of reduced lower limb can be confusing for the reader as you refer to lower limb biomechanics, and "gait" is a pattern of locomotion rather than an activity.

Line: 353: Consider rephrasing to remove the use of "likely" twice in the same sentence.

Line 407: Include a space between reference and text.

Kind regards.

Reviewer #2: This is an interesting study on the effects of simulated reduced gravity on walking of healthy young participants. In a factorial design, the effects of weight support and speed on a number of relevant variables are assessed. I have read the manuscript with great interest, the research question seems relevant, and the study seems well conducted. My main comments concern the analysis and discussion section. I have further specified my concerns below.

Main comments

The authors have chosen to vary weight support over a wide range. In addition, all participants walked at 4 different speeds, ultimately resulting in a factorial design with 16 cells. The assumption that underlies this design is that the effects of reduced gravity vary as a function of walking speed. However, the statistical analysis and, to some extent, the discussion section, focus mainly on the effects of reduced gravity rather than the interaction between reduced gravity and walking speed. Nevertheless, the figures seem to indicate that such interactions do exist for certain parameters. These interactions may be relevant e.g. for clinical application in patients with very slow walking speeds. So I wonder whether it would be useful to include the interactions in the statistical model and present the results in the manuscript (possibly as supplementary material?).

The discussion sometimes seems unnecessarily long. In certain passages, an extensive literature review is given with respect to a particular result without making much of a point about the data collected in the current study. I think the discussion can have more impact if it would be slightly more concise.

Introduction

Lines 62-65: It may be worth specifying here why and how walking speed may affect the effects of reduced gravity (and make clear why it would be interesting to study the interactions between both variables).

I appreciate that the authors take the trouble to formulate a number hypotheses. However, given the large number of variables, the hypotheses put forward seem somewhat arbitrary. As a reader I would have no qualms if this was presented as a descriptive study mapping (without specific hypotheses) the combined effects of speed and weight support .

Methods

Because participants had to step on force plates, were the participants instructed in a specific way prior to the measurement (for example, were they instructed to walk ‘as normally as possible’ or to land on the heel)? This seems particularly relevant as reduced gravity at higher walking speed appears to result in a more dorsiflexed landing (see fig 4), which seems a bit counterintuitive.

Line 144: Was there initial heel contact in all conditions, or did high levels of reduced gravity also result in forefoot landing? If the latter is the case, initial contact seems to be a better term here than heel strike.

Lines 161-164: Why were the muscle data low-pass filtered after the time normalization and averaging? It is common to do this on the rectified, raw signal.

Lines 173-175: Both speed and reduced gravity are included as factors in the statistical analysis. The main effects of speed (less relevant) and the interactions of speed with the level of gravity (certainly relevant) are not explicitly mentioned anywhere. Furthermore, it is not entirely clear to me what exactly the post hoc comparisons relate to. The different levels of the independent variable are not compared post hoc. Given the large number of dependent variables, some correction for multiple testing seems warranted, but I don't think these are ‘post hoc tests’.

Results

Line 181: 'some selected data points'. Is it possible to be a little more specific?

Lines 181-183: How was this determined?

I find the tables a bit confusing. The descriptive statistics are given for all variables in Table 1 and but not in Table 2. Perhaps the type of information provided in both tables can be it more consistent. With regard to Table 2: I don’t think you need to report the ranks and the results for the original test (before the Benjamini-Hochberg correction). Also, p-values of 0.000 are meaningless, please report p<.001.

Some figures seem to be slightly redundant (in particular figures 3, 7, and 9) as they contain information that is already presented in other figures.

Lines 251-255: I am not sure if this belongs in a Results section.

Lines 255-256: These and similar passages suggest that subphases of the gait cycle have been analysed, but this is not the case if I understand correctly.

Discussion

General: much of the discussion focusses on the EMG, and I think the relationship between the kinetic/kinematic/neuromuscular variables could be discussed more thoroughly. The effects on, for example, the braking and propulsive impulse are likely to alter neuromuscular control. Furthermore, it is perhaps good to discuss where / how reduced gravity does or does not result in a breakdown of the normal locomotor pattern. This can be particularly relevant for clinical applications when therapists strive to impose a 'physiological' gait pattern (under conditions of reduced gravity). A related relevant point is whether or not reduced gravity results in changes in the underlying locomotor task. Although weight support in patients is intended to simplify the walking task (by reducing task demands associated with weight gain and stability) anyone who has ever walked with a lot of weight support (e.g. > 50%) knows that this is hard and fatiguing.

General: at some points the discussion can be a bit more concise. E.g. the comparison with previous studies is relevant, but could perhaps be a bit more to-the-point. Some passages also seem to be a bit off-topic in the context of the present research question, e.g. a discussion of the reflex pathways.

Lines 282-284: I am not convinced that the Soleus and Lateral Gastrocnemius show no effect of reduced gravity. In particular at the end of the stance phase at high speeds, there seems to be an effect. Because the statistical analysis of the EMG is based on RMS values calculated over the entire gait cycle, these effects may not have been detected. However, they do become clear in Figure 8.

Lines 286-288: This seems like a strange hypothesis, as the quadriceps contribute to weight acceptance during the early stance phase. So it seems logical that with reduced gravity the amplitude of the activity in this muscle group decreases.

Lines 319-320: If you look at Figure 8, however, you can see that in the RF during the stance-swing transition, there appears to be an effect of reduced gravity.

Lines 352-354: Can the authors clarify this further? How (!) do differences in speed and the support system explain the discrepancies with previous studies that are observed?

Lines 356-358: I don't quite understand what the authors mean. Reformulate?

Lines 393-396: I find the suggestion interesting but perhaps the authors can further specify why / how conditions of reduced gravity changes the force-length relationship. Furthermore, the authors may be able to say something about why effects were in fact found in the MG.

Lines 416-420: I understand this is difficult to explain, but if differences between treadmill and overground are an explanation, this should be explained further: why? The effects are quite pronounced.

I do not find the paragraph on the clinical implications of the results very convincing. The main reason for having patients walk with reduced gravity (treadmill or overground) is to reduce problems in support and balance. In the context of motor (re-) learning it is important to know whether or not reductions in gravity result in deviating kinematic / kinetic / EMG patterns, in other words: does the weight support not introduce new task elements that are irrelevant for natural, overground walking, and does the support not result in a less ‘physiological’ gait pattern?

6. PLOS authors have the option to publish the peer review history of their article (what does this mean?). If published, this will include your full peer review and any attached files.

Reviewer #1: No

Reviewer #2: No

---

## [Author Response · Author response to Decision Letter 0]

25 May 2021

We have made revisions to address all the comments from the reviewers. A document is now included that has specific replies to each specific comment from the referees.

---

## [Decision Letter · Decision Letter 1]

7 Jun 2021

Human muscle activity and lower limb biomechanics of overground walking at varying levels of simulated reduced gravity and gait speeds

PONE-D-21-02036R1

Dear Dr. Ferris,

We’re pleased to inform you that your manuscript has been judged scientifically suitable for publication and will be formally accepted for publication once it meets all outstanding technical requirements.

Kind regards,

Pei-Chun Kao

Academic Editor

PLOS ONE

Additional Editor Comments (optional):

Reviewers' comments:

Reviewer's Responses to Questions

**Comments to the Author**

1. If the authors have adequately addressed your comments raised in a previous round of review and you feel that this manuscript is now acceptable for publication, you may indicate that here to bypass the “Comments to the Author” section, enter your conflict of interest statement in the “Confidential to Editor” section, and submit your "Accept" recommendation.

Reviewer #2: All comments have been addressed

2. Is the manuscript technically sound, and do the data support the conclusions?

Reviewer #2: Yes

3. Has the statistical analysis been performed appropriately and rigorously? 

Reviewer #2: Yes

4. Have the authors made all data underlying the findings in their manuscript fully available?

Reviewer #2: Yes

5. Is the manuscript presented in an intelligible fashion and written in standard English?

Reviewer #2: Yes

6. Review Comments to the Author

Reviewer #2: (No Response)

7. PLOS authors have the option to publish the peer review history of their article (what does this mean?). If published, this will include your full peer review and any attached files.

Reviewer #2: No

---

## [Editor Report · Acceptance letter]

18 Jun 2021

PONE-D-21-02036R1 

Human muscle activity and lower limb biomechanics of overground walking at varying levels of simulated reduced gravity and gait speeds 

Dear Dr. Ferris:

I'm pleased to inform you that your manuscript has been deemed suitable for publication in PLOS ONE. Congratulations! Your manuscript is now with our production department. 

Kind regards, 

on behalf of

Dr. Pei-Chun Kao 

Academic Editor

PLOS ONE